# Length of stay and economic sustainability of virtual ward care in a medium-sized hospital of the UK: a retrospective longitudinal study

Abdollah Jalilian,[1] Luigi Sedda  ,[1] Alison Unsworth,[2] Martin Farrier  [2]

[1]Lancaster Ecology and Epidemiology Group, Lancaster University, Lancaster, UK
[2]Wrightington Wigan and Leigh Teaching Hospitals NHS Foundation Trust, Wigan, UK

**Correspondence to**
Dr Luigi Sedda;
l.sedda@lancaster.ac.uk

## ABSTRACT

**Objective** To evaluate the length of stay difference and its economic implications between hospital patients and virtual ward patients.

**Design** Retrospective longitudinal study.

**Setting** Wrightington, Wigan and Leigh (WWL) Teaching Hospitals, National Health Service (NHS) Foundation Trust, a medium-sized NHS trust in the north-west of England.

**Participants** Virtual ward patients (n=318) were matched 1:1 to 1:4, depending on matching characteristics, to all hospital patients (n=350). All patients were admitted to the hospital during the calendar year 2022.

**Outcome measures** The primary outcome is the length of stay as defined from the date of hospital admission to the date of discharge or death (hospital patients) and from the date of hospital admission to the date of admission in a virtual ward (virtual ward patients). The secondary outcome is the cost of a hospital bed day and the equivalent value of virtual ward savings in hospital bed days. Additional measures were 6-month readmission rates and survival rates at the follow-up date of 30 April 2023.

**Risk factors** Age, sex, comorbidities and the clinical frailty score (CFS) were used to evaluate the importance and effect of these factors on the main and secondary outcomes.

**Methods** Statistical analyses included logistic and binomial mixed models for the length of stay in the hospital and readmission rate outcomes, as well as a Cox proportional hazard model for the survival of the patients.

**Results** The virtual ward patients had a shorter stay in the hospital before being admitted to the virtual ward (2.89 days, 95% CI 2.1 to 3.9 days). Chronic kidney disease (CKD) and frailty were associated with a longer length of stay in the hospital (58%, 95% CI 22% to 100%) compared with patients without CKD, and 14% (95% CI 8% to 21%) compared with patients with one unit lower CFS. The frailty score was also associated with a higher rate of readmission within 6 months and lower survival. Being admitted to the virtual ward slightly improved survival, although when readmitted, survival deteriorated rapidly. The cost of a 24-hour period in a general hospital bed is £536. The cost of a day hospital saved by a virtual ward was £935.

**Conclusion** The use of a 40-bed virtual ward was clinically effective in terms of survival for patients not needing readmission and allowed for the freeing of three

## STRENGTHS AND LIMITATIONS OF THIS STUDY

⇒ The largest virtual wards study, in terms of virtual ward patients, and the first in the UK assessing their cost.
⇒ The information about inclusion and exclusion criteria used to send patients to virtual wards was subjective and therefore subject to selection bias.
⇒ Exact costs for hospital patients were not available for this research, causing some uncertainty about the true cost difference between hospital and virtual ward patients.
⇒ Given the sample size, only a few individual factors were used for matching; therefore, not all potential confounders could have been considered.

hospital beds per day. However, the cost for each day freed from hospital stay was three-quarters larger than the one for a single-day hospital bed. This raises concerns about the deployment of large-scale virtual wards without the existence of policies and plans for their cost-effective management.

## INTRODUCTION

Virtual wards offer patients the opportunity to receive healthcare in the comfort and safety of their own homes, including care homes.[1] Pioneered in the UK, virtual wards became a popular scheme during COVID-19[2] and since then it has been replicated in other 'thematic virtual wards' (ie, respiratory, heart failure (HF) and chronic obstructive pulmonary disease (COPD), to name a few). Virtual wards are the response to the chronic and unsustainable high demand for secondary care beds.[3] By providing an alternative to continued admission, virtual wards may help bridge the gap between demand and capacity for hospital beds, especially in the context of an ageing population,[3 4] although their successes are still debated (see, eg, Creavin and colleagues' review of randomised controlled trials of virtual wards for acute respiratory infection[5]). A plan to enrol in

virtual wards in most of the integrated care systems in the UK was launched in 2020 with the aim of making available hundreds of thousands of virtual beds (at a cost of hundreds of millions of pounds) in the next few years in order to provide them as an option for every patient.[6]

Virtual wards must be cost-effective if they are to replace traditional inpatient care; the costs must be comparable or lower than the costs of hospital stays to be economically sustainable in the medium to long term. To establish whether large-scale implementation of virtual wards could alter the provision of healthcare, it is necessary therefore to address workforce issues and reduce the cost of modern healthcare.

The main objective of this research is to compare the length of stay in a medium-sized UK hospital for hospital and virtual ward admitted patients. We define the length of stay as the duration, in days, between the date of admission to the hospital and the date of discharge from the hospital for hospital patients and from the date of admission to the hospital and the date of admission to the virtual ward for virtual ward patients. By analysing the length of stay, we aim to assess the effectiveness of virtual wards in reducing the duration of hospitalisation and therefore increasing the number of beds available for hospital patients. This is also defined as the 'step-down' model.[3]

Beyond potential reductions in the length of a patient's stay in the hospital, we aim to compare the survival outcomes of patients in both virtual wards and hospital settings. Additionally, we compare the readmission rates to the hospital within 6 months of discharge. Survival and readmission are two indicators of the clinical effectiveness of the virtual ward in satisfying the healthcare needs of the patients.[7]

By considering all these aspects—length of stay, survival, readmission rates and costs—it will be possible to assess the impact of virtual wards on patient care and their effectiveness in comparison to traditional inpatient care. The findings from this study respond to recent requests for evidence-based, informed virtual ward provision[8] and will contribute to a deeper understanding of the potential benefits and implications of virtual wards as an alternative care model.

## MATERIALS AND METHODS
### Virtual ward
The virtual ward at Wrightington, Wigan and Leigh (WWL) Teaching Hospitals National Health Service (NHS) Foundation trust initially originated as an oxygen service to facilitate the follow-up of COVID-19 patients. As the number of COVID-19 cases requiring follow-up decreased, the service underwent repurposing. The administration of virtual ward patient management used a software platform provided by Current Health, with monitoring equipment established by the WWL medical electronics team to generate automated data through the Current Health web-based system. By the onset of 2022, the virtual ward had evolved into a stable service, providing a viable alternative to continuous hospital admissions.

A dedicated team of core nurses maintained daily communication with patients through iPad and Face-Time. Patient data are systematically scrutinised through the analysis of automated observations within a cloud-based software framework. In instances where concerns arise about a patient's condition, virtual ward nurses coordinate a review with the responsible consultant. In-person visits for tasks such as intravenous antibiotic administration, blood sample collection or wound management were organised by the nurses. Continuous monitoring is conducted to track the patient's progress or detect signs of deterioration. In cases of deterioration, arrangements for readmission can be made, with transportation facilitated to the emergency department or same-day emergency care through ambulance services if necessary.

The selection of patients for the virtual ward is a collaborative effort between the medical and nursing teams on the ward. The virtual ward team conducts assessments to determine suitability, and on confirmation, a technical setup visit is arranged at the patient's residence to verify self-sufficiency, assistance from a partner or family or an existing care support system. Patients relocated to care homes are excluded from virtual ward consideration.

Discharge from the virtual ward is contingent on assessments conducted by the virtual ward team and only when monitoring is deemed unnecessary for the patient.

### Data
The study follows a retrospective longitudinal design where virtual ward patients undergo a partial discharge process from the hospital and continue to receive medical attention and support within their homes until they complete their treatment and are then fully discharged; and hospital patients receive care within the hospital until they are fit for full discharge.

Virtual ward patients are matched manually with patients admitted exclusively to the hospital, resulting in 350 virtual ward hospital pairs. Patient Administration System data were used to pull a list of age-matched patients who were admitted during the same study year. Further data from coding were then used to provide full matching for virtual patients. These patient pairs comprised 350 unique hospital patients and 318 virtual ward patients. Therefore, 291 are matched to one virtual ward to one hospital patient (1:1); 23 are 1:2; three are 1:3 and finally one is 1:4. The study group comprises all the patients admitted to WWL Teaching Hospitals NHS Foundation during the calendar year 2022 (1 January 2022 to the 31 December 2022) and their progress was followed up on the 30 April 2023.

Matching characteristics were sex, age, primary diagnosis description and clinical frailty score (CFS, 1 representing a very fit patient to 9 indicating a terminally ill patient). CFS is measured at triage by the care professional seeing the patient (it may also be reassessed at a

later point in the admission by other healthcare professionals if further information becomes available). Attribution of one of the CFS to a patient was carried with the support of the Clinical Frailty Scale App.[9]

All matched patients have the same gender and the same primary diagnosis description. Ninety-seven per cent of the matched patients possess the same CFS, while the remaining 3% have CFSs within a range of ±2 points. Finally, 95% of the matched patients share the same age or have an age difference within ±6 years, with the remaining 5% having an age gap of less than 12 years. Matching was exact where possible, but for a minority of patients, an exact match was not possible in the dataset for the study year. In that case, the closest possible match was achieved, leading to the ranges in CFS and age.

Additional unmatched information was provided for each patient: comorbidities, readmission information, date of admission and discharge to and from the hospital, date of admission and discharge to and from the virtual ward and date of death, when applicable, up to 30 April 2023. International Classification of Diseases, 10th revision-based coding data were employed to assign comorbidities to each patient. Each patient's episodes of care were meticulously coded at the point of discharge, offering comprehensive insights into both the reason for admission and the documented comorbidities. These comorbidities were then cross-referenced with the records obtained at the time of admission to the virtual ward. In instances where disparities were identified between the recorded comorbidities at admission and those during the episodes of care, the electronic notes of each patient were meticulously examined to establish an accurate set of comorbidities. The electronic notes served as a reference in resolving any discrepancies, ensuring the utmost precision in our data interpretation. Due to the study design and rigorous data entry checks, there is no missing data.

Forty-eight patients admitted to virtual wards did not have suitable matches among the patients admitted exclusively to the hospital during the study year. Consequently, these 48 patients were excluded from the analyses.

To estimate the cost-effectiveness of virtual ward admission compared with exclusively hospital ward admission, the costs of facilities and staffing were included in the analysis. Financial data for the cost of the virtual ward was maintained separately from other trust finances so that the cost of the virtual ward could be fully understood. Financial data were provided by the WWL Trust Finance Team.

### Patient and public involvement
Owing to the use of retrospective data sources, no members of the public or patients were formally involved in the study's design, analysis or manuscript writing due to the limited size of the research group and the absence of funding. The communication plan includes executive summaries circulated at the NHS trust, press releases, public lectures and social media posts to reach the most diverse audience.

### Methods
Study objectives and respective methods are summarised in table 1. Because some virtual ward patients have been matched with more than one hospital patient, a random effect representing recurring virtual ward patients is included in the models.

Models were run within a Bayesian framework with assigned non-informative prior distributions for all parameters. The computations were performed using the R-Integrated Nested Laplace Approximation (INLA) package[10] to allow to incorporate uncertainty into the model and estimate the posterior distributions of the model parameters using the INLA methodology.[10]

**Table 1** Statistical analyses employed for the four study outcomes

|  | **Primary outcome** | **Secondary outcome** | **Additional outcomes** | |
|---|---|---|---|---|
|  | Length of stay (in days) | Bed cost | Being readmitted to the hospital within 6 months after discharge.[21] | Survival at the follow-up date of 30 April 2023 |
| Question | Which are the factors influencing the length of stay in the hospital? | What is the daily cost of a bed in a hospital and a bed in a virtual ward? | What are the factors associated with the readmission of patients within 6 months after discharge? | What are the factors contributing to the survival of patients? |
| Statistical model | Generalised mixed model with generalised Poisson response | Descriptive approach: direct collation of cost for virtual ward compared with National Health Service-approved methodology for non-elective inpatient bed costs at the same organisation. | Logistic regression model | Cox proportional hazard model |

The virtual ward cost is calculated from the accrued expenditure. This was kept as a separate spreadsheet so that the exact costs were known. The virtual ward uses 12.5 whole-time equivalent (WTE) nurses and 0.5 WTE consultants with the assistance of two WTE pharmacists and 6.8 WTE ancillary staff members. National Institute for Health Care and Excellence recommendations[11] provide a basis for the calculation of standard ward staffing for an idealised inpatient ward. Using their recommendations, a typical 28-bed hospital ward would require 35 WTE nurses to staff with 12 WTE ancillary staff members. Medical staffing for a traditional 28-bed ward would depend on the clinical work but would require a ward round each day and the involvement of a consultant and junior medical staff.

Cost estimation for hospital bed days uses NHS-agreed methodology. It represents the average cost of non-elective care provision in wards on the acute hospital site. It is calculated by the Finance Department at the organisation and is in line with the costs submitted by the organisation to NHS England.

Patients who are cared for in the virtual ward are, by definition, in the second half of their inpatient hospital stay. For hospital patients, costs are higher for the first day of a patient's admission to the hospital than they are for subsequent days because patients are likely to need more care, tests and intervention in the first part of their stay. At the moment, an estimate of the cost for the second half of a patient's stay in the hospital is not available; only the average cost for all days spent in the hospital has been provided. As a consequence, the cost estimate of hospital stay is a high estimate, and the virtual ward cost is an exact cost.

## RESULTS
### Summative and descriptive analysis
The primary diagnosis description for the majority (64.3%) of the matched patients corresponds to the conditions pneumonia, COPD, COVID-19, atrial fibrillation or HF. Overall, patients admitted to virtual wards exhibit a shorter hospital stay in comparison to their matched counterparts in the hospital (p<0.001) (table 2). Based on a 95% CI, it is estimated that virtual ward patients experience a reduction in hospital stay ranging from 2.1 to 3.9 days compared with their corresponding matches in the hospital.

In general, patients admitted to virtual wards have lower prevalence rates of asthma, HF, chronic kidney disease (CKD), COPD, diabetes, hypertension and learning disability compared with the patients staying in hospitals. Moreover, a slightly higher percentage of patients admitted to virtual wards were alive on the follow-up date in comparison to patients admitted exclusively to the hospital (although bordering statistical significance). Finally, the readmission rate within 6 months after discharge is higher for patients admitted to virtual wards

than hospital patients. Only one patient was readmitted to the hospital during the virtual ward clinic.

Virtual ward patients used to match hospital patients were representative of the total virtual ward patient cohort (online supplemental table S1).

### Length of stay in hospital, readmission and survival models
The results of the logistic regression model to the binary variable of being readmitted within 6 months of discharge (summarised in table 3) indicate that several factors are significantly influencing the likelihood of readmission within 6 months of discharge. Older age, higher CFS and having COPD are associated with increased chances of readmission. Specifically, for every one-unit increase in age, while keeping all other factors at their average level, there is a 3% increase in the odds of readmission. Similarly, for each unit increase in the CFS, while holding all other factors at their average level, the odds of readmission increase by 20%. Having a diagnosis of COPD leads to a 70% higher odds of readmission. Notably, being admitted to virtual wards is the most substantial risk factor for readmission within the next 6 months. Virtual ward patients face a significant 120% increased odds of readmission compared with their counterparts in the hospital.

The results from the Cox proportional hazard model applied to the survival from admission to either death or being alive at the follow-up date are shown in table 4. The proportional hazards assumption for the fitted Cox model was validated (p=0.11), indicating that the hazards were proportional over time. Based on these results, the following conclusions can be made:
1. Patients admitted to the virtual ward had a significantly lower mortality (63%) compared with those admitted to the hospital.
2. Patients with heart failure show a 75% higher risk of death compared with those without this condition.
3. For each unit increase in the CFS, while holding all other factors at their average level, the risk of death increases by approximately 35%.
4. There is a 4% increase in the risk of death for each additional day of hospital stay while keeping all other factors at their average level.
5. Patients who are readmitted within 6 months of discharge face a significant increase in the risk of death. Specifically, virtual ward patients readmitted to hospital experience a 220% higher risk of dying compared with the 175% of hospital patients readmitted to hospital when compared with non-readmission patients.

Finally, the length of stay is associated with several factors (table 5):
1. Having CKD is associated with an average increase of around 60% in the length of stay.
2. For each unit increase in the CFS, while holding all other factors at their average level, there is an average increase of around 15% in the length of stay.
3. Being admitted to virtual wards, as opposed to staying in the hospital, results in an average decrease of

**Table 2** Baseline demographics of virtual ward and hospital patients

| Baseline demographics | Patients admitted only to the hospital (n=350) | Patients admitted to virtual wards (n=318) | P value |
|---|---|---|---|
| Percentage of female patients | 54.6% | 54.7% | |
| Median age (years) | 72 | 71 | |
| Median Clinical Frailty Score | 4 | 4 | |
| Number (percentage) of patients with pneumonia as their primary diagnosis description | 70 (20%) | 61 (19.2%) | |
| Number (percentage) of patients with chronic obstructive pulmonary disease as their primary diagnosis description | 63 (18%) | 55 (17.3%) | |
| Number (percentage) of patients with COVID-19 as their primary diagnosis description | 38 (10.9%) | 33 (10.4%) | |
| Number (percentage) of patients with atrial fibrillation as their primary diagnosis description | 30 (8.6%) | 27 (8.5%) | |
| Number (percentage) of patients with heart failure as their primary diagnosis description | 24 (6.9%) | 22 (6.9%) | |
| Number (percentage) of patients with hypertension as their primary diagnosis description | 15 (4.3%) | 14 (4.4%) | |
| Number (percentage) of patients with cellulitis as their primary diagnosis description | 11 (3.1%) | 11 (3.4%) | |
| Number (percentage) of patients with chest infection as their primary diagnosis description | 10 (2.9%) | 10 (3.1%) | |
| Average length of stay in the hospital (from hospital admission to hospital discharge or to virtual ward) | 5.96 days | 2.89 days | *** |
| Average length of stay in virtual wards (from virtual wards onboarding to virtual wards offboarding) | – | 9.79 days | NA |
| Total number of hospital days used by patients | 2086 | 920 | |
| Total number of virtual wards days used by patients | – | 3114 | |
| Average length of care in both hospital and virtual wards | 5.96 days | 12.96 days | *** |
| Number (percentage) of patients with diabetes | 86 (24.6%) | 47 (14.8%) | ** |
| Number (percentage) of patients with hypertension | 159 (45.4%) | 103 (32.4%) | *** |
| Number (percentage) of patients with chronic heart disease | 123 (35.1%) | 36 (11.3%) | *** |
| Number (percentage) of patients with chronic kidney disease | 51 (14.6%) | 2 (0.6%) | *** |
| Number (percentage) of patients with chronic obstructive pulmonary disease | 128 (36.6%) | 81 (25.5%) | ** |
| Number (percentage) of patients with asthma | 39 (11.1%) | 8 (2.5%) | *** |
| Number (percentage) of patients with learning disability | 5 (1.4%) | 1 (0.3%) | |
| Number (percentage) of patients being readmitted to the hospital within 6 months after discharge | 108 (30.9%) | 128 (40.3%) | * |
| Number (percentage) of patients being alive on the follow-up date of 30 April 2023 | 275 (72.4%) | 275 (86.5%) | * |

Number of stars resulting from the p-value obtained from a paired t-test: *if $0.01 \leq$ p-value$<0.05$; **if $0.001 \leq$ p-value$<0.01$ and ***if p-value$<0.001$. For comorbidities, only those above 2% are shown.

around 45% in the length of stay in the hospital before admission to a virtual ward.

Including an interaction term for each comorbidity and hospital treatment (virtual ward or inpatient hospital) does not produce statistically significant coefficients for any of these terms.

### Hospital and virtual ward costs

The costs for the virtual ward during the calendar year 2022 are shown in online supplemental table S2. The total was around £1.051 million for 40 virtual ward beds. There were additional costs of £100 000 as part of the process of setting up the service, but these are non-recurrent costs

**Table 3** ORs and their corresponding 95% credible intervals for factors affecting the probability of being readmitted within 6 months of attendance (in bold, the statistically significant factors at a significance level of 5%)

|  | ORs | 95% credible interval | |
| --- | --- | --- | --- |
| Intercept | 0.033 | 0.01 | 0.083 |
| Gender, male | 0.756 | 0.531 | 1.044 |
| Age | **1.027** | **1.011** | **1.043** |
| Clinical Frailty Score | **1.176** | **1.035** | **1.329** |
| Diabetes | 1.441 | 0.924 | 2.146 |
| Hypertension | 0.732 | 0.508 | 1.022 |
| Heart failure | 0.955 | 0.61 | 1.424 |
| Chronic kidney disease | 1.408 | 0.7 | 2.54 |
| Chronic obstructive pulmonary disease | **1.673** | **1.148** | **2.356** |
| Asthma | 0.949 | 0.428 | 1.833 |
| Learning disability | 9.104 | 0.937 | 36.551 |
| Length of stay | 1.023 | 0.997 | 1.048 |
| Admitted to virtual wards | **2.226** | **1.506** | **3.172** |

and therefore not included in the final costings for the virtual ward.

The calculated reduction of inpatient hospital bed days for patients admitted to the virtual ward was 3.07 days for

**Table 4** HRs and their corresponding 95% credible intervals for factors affecting the death of patients from the date of admission (in bold, the statistically significant factors at a significance level of 5%)

|  | HR | 95% credible interval | |
| --- | --- | --- | --- |
| Intercept | 0.001 | 0 | 0.004 |
| Gender, male | 1.03 | 0.703 | 1.457 |
| Age | 1.009 | 0.99 | 1.028 |
| Clinical Frailty Score | **1.351** | **1.181** | **1.538** |
| Diabetes | 0.966 | 0.605 | 1.465 |
| Hypertension | 0.915 | 0.615 | 1.311 |
| Heart failure | **1.762** | **1.143** | **2.597** |
| Chronic kidney disease | 1.047 | 0.556 | 1.8 |
| Chronic obstructive pulmonary disease | 1.376 | 0.937 | 1.951 |
| Asthma | 0.405 | 0.104 | 1.097 |
| Learning disability | 0.867 | 0.065 | 3.843 |
| Admitted to virtual ward | **0.370** | **0.160** | **0.750** |
| Length of stay | **1.04** | **1.02** | **1.06** |
| Readmission 6 months: hospital patient | **2.763** | **1.688** | **4.272** |
| Readmission 6 months: admitted to virtual wards | **3.179** | **2** | **4.805** |

**Table 5** ORs and their corresponding 95% credible intervals for factors associated to the length of stay in hospital of patients (in bold, the statistically significant factors at a significance level of 5%)

|  | ORs | 95% credible interval | |
| --- | --- | --- | --- |
| Intercept | 4.116 | 2.566 | 6.29 |
| Gender, male | 0.938 | 0.796 | 1.099 |
| Age | 0.995 | 0.988 | 1.002 |
| Clinical Frailty Score | **1.144** | **1.077** | **1.213** |
| Diabetes | 1.103 | 0.91 | 1.323 |
| Hypertension | 1.035 | 0.874 | 1.217 |
| Heart failure | 1.134 | 0.918 | 1.383 |
| Chronic kidney disease | **1.579** | **1.224** | **2.004** |
| Chronic obstructive pulmonary disease | 1.077 | 0.872 | 1.315 |
| Asthma | 1.163 | 0.867 | 1.527 |
| Learning disability | 1.287 | 0.651 | 2.29 |
| Readmission within 6 months | 1.084 | 0.918 | 1.271 |
| Admitting to virtual wards | **0.563** | **0.47** | **0.669** |

each of the patients admitted to the virtual ward (table 2, 5.96 vs 2.89 days). There were 366 patients admitted to the virtual ward during the study period, for which the costs of the virtual ward were £1.051 million. Using those figures, the cost of saving one inpatient hospital day was £935 (£1 051 150/ (3.07 days×366 patients)).

The average cost of a non-elective inpatient bed day within the organisation using standard NHS methodology was provided by the WWL Finance Department and amounted to £536, a cost coherent with other studies.[12]

A comparison of costs for the virtual ward with the costs of a continued inpatient stay is therefore possible. The cost of reducing a patient's hospital stay by 1 day is £935 when they are admitted to the virtual ward as a strategy for shortening their inpatient stay. If the patient had stayed in the hospital rather than being admitted to the virtual ward, the equivalent inpatient hospital day cost would have been £536. Therefore, the cost of virtual ward care is approximately three-quarters higher than that of traditional inpatient care. However, the £935 is calculated based on the WWL's capacity to use the virtual beds, which was 24% of the potential 14 600 (24-hour) beds per year provided by 40 virtual ward beds (online supplemental table S1, reference to 3508 total days spent by virtual ward patients).

## DISCUSSION

This study has shown that virtual ward patients had a shorter length of stay in hospital but were more likely to be readmitted, and when readmitted, had a lower chance to survive compared with non-readmitted patients and

readmitted patients previously discharged from hospital. Virtual ward patients were affected by a lower number of comorbidities, although their frailty scores along with sex, age and primary diagnosis were matched with hospital patients. While virtual wards are shown to effectively reduce the length of inpatient stays, the cost associated with a freed-day hospital bed is three-quarters higher than that of inpatient hospital care.

Previous studies have focused on the development of virtual wards for specific conditions or groups of patients. For the UK, one of the first countries to trial virtual wards, literature is scarce and the size of virtual wards is smaller compared with the one presented here.[3 13 14] In addition, this study is the first example of the assessment of virtual ward cost-effectiveness in the UK within a medium-size hospital facility.

In terms of virtual ward clinical outcomes, current knowledge provides conflicting evidence,[15] apart from randomised controlled trials enrolling patients with HF.[7] Although a direct comparison of the present study with the current literature on clinical outcomes is not possible, some common patterns can be highlighted.

As in other studies, patients admitted to the virtual ward had lower rates of comorbidity (see, eg, the review for the virtual ward COVID-19 by Majoor and Vorselaars[16]). While recent literature reviews found a lower rate of readmission in older patients with COPD[3] and COVID-19 patients[17] after hospitalisation at home and virtual wards, the present study did not find any significant difference, as also reported in Creavin *et al*.[5] However, readmission rates for all patients (hospital and virtual wards) increased if a patient was from a virtual ward, older and had COPD. The higher rate of readmission is coherent with clinical expectations. The care provided by the virtual ward is, in part, a monitoring process; therefore, clinical deterioration is often followed by readmission of the patient.[13] For discharged hospital patients, readmission would need to be initiated by the patient. For those virtual ward patients not readmitted, the care is shown to be non-inferior, and other studies have indicated that care in virtual wards is preferred by patients and also by the healthcare staff.[3 18–20]

Survival was better in patients admitted to the virtual ward but decreased in patients from virtual wards when readmitted and for those with HF. A similar result was found in other studies.[3 21] Although virtual ward patients were matched by sex, age, primary diagnosis description and CFS, they showed a lower frequency of comorbidities, which may have contributed to a larger survival rate on their first admission when compared with hospital patients. It is likely that, despite the careful matching of patients, there is a selection bias towards fitter patients when patients are being selected for consideration of admission to the virtual ward. Greater independence and the ability to communicate with virtual ward staff by videophone make these patients better candidates for admission to the virtual ward.

Our analysis found the length of stay in hospital prior to virtual ward admission to be shorter than the one in other studies, although many of them were based on COVID-19 virtual wards or other disease-specific virtual wards.[5 19 22–24] A common result, however, is a shorter stay in the hospital for virtual ward patients than hospital patients. The present study highlighted that, apart from CKD, the reduction in length of stay is not related to any other primary diagnosis, although it is dependent on the assessed CFS. Length of stay increases in patients with higher CFS and/or CKD.

It is important to note that CFS is detrimental to all the clinical outcomes considered in this analysis: length of stay, readmission and survival. Another study, although on an elderly cohort, showed a similar association between CFS and mortality.[25] These findings may suggest that patients with lower CFS are those who may benefit more from virtual wards in terms of cost-effectiveness and clinical outcome. Clinical experience indicates that virtual ward care requires a patient to have some self-care ability. The more compromised that is, the less likely a virtual ward is to be able to provide appropriate levels of care.

The exclusion of comorbidity indexes, such as the Charlson[26] or Elixhauser,[27] in our study, was driven by two considerations. First, our research focused on the analysis of length of stay, survival and readmission at the level of individual comorbidities rather than a broader assessment of general comorbidity status. Second, this approach allowed for a more detailed examination of specific health conditions and facilitated comparisons with findings from prior research that adopted a similar methodology. However, it is crucial to recognise that in any future research aimed at optimising the decision-making process regarding patients' admission to hospitals or virtual wards, an exploration of the discriminatory power of established comorbidity indexes such as Charlson or Elixhauser would be highly beneficial.

With a focus on the length of stay, which is the key aspect for the target of increasing hospital bed provision by the NHS in the UK,[28] this retrospective longitudinal study of a virtual ward through a whole calendar year provides evidence of the shorter length of stay in hospital for patients sent to virtual wards compared with hospital inpatients. Virtual ward leads to a reduction in length of stay from 5.96 days to 2.89 days (a reduction of 3.07 days) on average, one of the shortest in the current literature as described above. The reduction in length of stay is achieved without an increase in the mortality of the virtual ward patients, although higher readmission rates and lower survival for virtual ward patients readmitted to the hospital were found. Patients with no HF or CKD had a shorter length of stay compared with the rest of the cohort.

The reduction in length of stay is an average of 3.07 days for patients cared for by the virtual ward in 2022. Considering this cost was based on 366 virtual ward patients, this equates to 1123.62 hospital bed days freed in a year (3.07×366). This means that for a hospital with 100% occupancy (therefore providing 365 hospital bed days per bed) and 40 virtual ward beds at the capacity found

in this study (24%), the virtual ward effect is an increase in the hospital capacity of 3.08 hospital beds per day (1123.62/365). The virtual ward saving of 3.08 hospital beds is around 1/9 of a whole typical 28-bedded ward. Therefore, 360 virtual ward beds are necessary to replace 28 hospital beds. The provision of the 360 virtual wards would be an extensive undertaking, and it is assumed that within the trust, there are patients who could appropriately be admitted to the 360 virtual ward beds. However, increasing the capacity to use the 40 virtual ward beds and reducing the time from hospital admission to virtual ward admission can reduce the cost of a freed-day hospital bed. Given the current virtual ward organisation, it is therefore unlikely that the virtual ward could replace a whole hospital ward using the virtual ward model as it was studied in this paper. Nor is it more cost-effective than traditional inpatient care. The original funding award agreed for the virtual ward was £4.4 million and would allow for a virtual ward of 160 virtual beds. That funding, on the basis of this study, would provide a replacement for approximately 12.5 hospital beds. But if the throughput of the virtual ward were to be doubled, then the virtual ward could replace almost a whole inpatient ward. The virtual ward would have 160 beds and would be staffed by 50 nurses and 28 additional healthcare assistant (HCA)/ancillary staff—significantly greater than the requirement of the equivalent inpatient ward (35 WTE nurses and 12 WTE HCA/ancillary staff).

If the total cost could have covered the potential 14 600 bed days from 40 virtual ward beds, then the cost for a virtual bed per day was around £72, a cost comparable to others reported in the literature (ie, €88 for virtual wards designed for severe chronic respiratory diseases[24]). However, the actual cost was £299 for each of the 3508 bed days. In total, the cost for a patient in-hospital was £3194 on average and that for a virtual ward patient (including the time spent in hospital) was £4336 on average. This is in contrast with some studies that have reported savings of up to €8000 from virtual wards, although the studies were of low quality[3 7] and not comparable to current UK NHS provision.

While the virtual ward is shown to be an effective way to reduce hospital stays, it requires more carers and is more expensive than a traditional hospital stay. To be cost-effective, the virtual ward would need to double its throughput without altering the standard of care or type of patients cared for. At that level, the service would be cost-effective but would still not provide a saving on traditional inpatient care. Virtual ward care has more readmissions and looks after patients with fewer comorbidities. To save on traditional care, it would need to aspire to triple throughput. Without further efficiency in the virtual ward provision,[29] it remains a clinically effective way to care for patients but not a cost-effective or efficient way to care for patients.

This retrospective, longitudinal study has several limitations. First, the relatively limited sample size combined with the manual matching process certainly introduces the potential for selection bias, as the matching may not comprehensively account for all relevant variables. In addition, including variations in matching ratios and the inability to find suitable matches for some virtual ward patients may have introduced variability in the comparison groups. For the cost analysis, while efforts are made to estimate costs associated with virtual ward and hospital admissions, variations in staffing levels, assumptions made in cost calculations and the unavailability of estimates for the second half of a patient's hospital stay introduce uncertainties into the accuracy of the cost estimates and therefore limit the completeness and generalisability of the cost findings.

**Acknowledgements** We thank the WWL financial and healthcare teams for supporting this work, and all the WWL patients, which data contributed to the findings of this work.

**Contributors** All authors conceived and designed the study. AU and MF collected and screened the data. AJ and LS analysed and interpreted the data. AJ and LS drafted the first version of the manuscript. All authors critically revised the manuscript for important intellectual content. MF and LS supervised the study. AU and MF had full access to all the data in the study and took responsibility for the integrity of the data and the accuracy of the data analysis. MF is the guarantor. The corresponding author attests that all listed authors meet authorship criteria and that no others meeting the criteria have been omitted. MF is a senior author.

**Funding** LS is supported by the NIHR Health and Social Care Delivery Research (HSDR) programme (NIHR134540) and North West Cancer Research UK (LI2021SEDDA). The views expressed are those of the author(s) and not necessarily those of the NIHR or the Department of Health and Social Care.

**Competing interests** LS's institution received consulting fees from WWL for the submitted work. MF is the Associate Medical Director at Wrightington, Wigan and Leigh Teaching Hospitals. AU and AJ declare no conflict of interest.

**Patient and public involvement** Patients and/or the public were not involved in the design, or conduct, or reporting, or dissemination plans of this research.

**Patient consent for publication** Not applicable.

**Ethics approval** Ethics application for use of anonymised secondary data has been approved by the Faculty of Health and Medicine Ethics Committee on the 13th of June 2023 (FHM REC Reference: FHM-2023-3699-DataOnly-3. Title: Evaluation of virtual wards in a medium size hospital in UK).

**Provenance and peer review** Not commissioned; externally peer reviewed.

**Data availability statement** Data are available upon reasonable request. Data are available on reasonable request from martin.farrier@wwl.nhs.uk. All data were anonymised, and data sharing followed the Trust Governance protocols and procedures. Identifiable patient-level data from this project are not available to the public.

**ORCID iDs**
Luigi Sedda http://orcid.org/0000-0002-9271-6596

Martin Farrier http://orcid.org/0000-0002-7556-6848

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
