## [Reviewer comments · BMJ Open]

ARTICLE DETAILS

TITLE (PROVISIONAL)	Length of stay and economical sustainability of virtual ward care in a medium sized hospital of the UK. A retrospective longitudinal study
AUTHORS	Jalilian, Abdollah; Sedda , Luigi; Unsworth, Alison; Farrier, Martin

VERSION 1 – REVIEW

REVIEWER	Hakendorf, P Flinders University
REVIEW RETURNED	23-Nov-2023

GENERAL COMMENTS	Thank you for the opportunity to review this well written, interesting, informative and important paper. There is a lot of work in hospitals around the world trying to optimise patient flow to reduce length of stay and improve hospital capacity. The demand for limited bed availability is increasing. This results in pressure to decrease LOS to improve hospital capacity and make more beds available. Some of the methods include reducing inpatient hospital length of stay by early discharge home to or to other services that do not require patients to occupy acute inpatient beds. These novel programs need to make sure that patients do not require subsequent readmission or worse still early death. This paper describes such a program that uses the concept of a virtual ward as an “alternative care model” by using the patients home with support as a type of stepdown facility I commend the authors on describing such a program which highlights the benefits but also describe some unexpected pitfalls such as increased readmissions that provide an area for study and improvements into the future. A few minor questions/queries Can you expand your definition of the virtual ward and generally explain how patients are eligible for discharge/transfer to the virtual ward and how they are treated/followed in this ward. Do they have regular visits from clinical staff during their time in the virtual ward and how this type of care differs from what other jurisdictions may call hospital in the home. Is the decision to discharge to a virtual ward influenced by the availability of other government or non government support services it easier to discharge to a Virtual ward if the patient has a partner or carer than those without. You mention that its easier to discharge to virtual care if the patient can self-care but is the availability of other carers such as a partner or other home services or patients residing in a nursing home able to provide care if a patient cannot self-care
---

	Could you define or show show a reference to how the clinical frailty score is calculated is it based on ICD codes or a clinical assessment team that estimated the score How were comorbidities measured ie by scanning clinical notes or looking at secondary icd code? Would a calculated index such as Charlson or Elixhauser have been useful to describe comorbidities
--	--

REVIEWER	Li, Chao-Jui Chang Gung Memorial Hospital Kaohsiung Branch, Department of Emergency Medicine
REVIEW RETURNED	24-Nov-2023

GENERAL COMMENTS	A case-control study is an observational research design used in epidemiology to investigate and compare individuals with a specific outcome or condition (cases) and individuals without that outcome or condition (controls). The primary aim is to identify factors or exposures that may be associated with the occurrence of the outcome. In a case-control study, researchers select cases based on the presence of the outcome of interest and then choose controls without the outcome, often from the same population. The two groups are then compared with respect to various exposures or risk factors to assess their association with the outcome. Matching criteria, such as age, gender, and other relevant variables, may be employed to control for potential confounding variables and enhance the validity of the findings. While this study performed substantial matching on various risk factors, the initial grouping was not based on a specific outcome but rather on differences in hospitalization patterns. I believe this does not align with the definition of a case-control study and is more akin to a retrospective cohort study In addition, in the section on experimental design, there is insufficient description of the care differences experienced by patients admitted to the hospital and those admitted to the virtual ward. I believe a more detailed explanation should be provided. Based on the above two problems, I believe the manuscript should undergo a major revision.
--

VERSION 1 – AUTHOR RESPONSE

Reviewer: 1

Dr. P Hakendorf, Flinders University

I commend the authors on describing such a program which highlights the benefits but also describe some unexpected pitfalls such as increased readmissions that provide an area for study and improvements into the future.

We thank Dr Hakendorf for their positive assessment of the work and the important questions they raised.

R1

Can you expand your definition of the virtual ward and generally explain how patients are eligible for discharge/transfer to the virtual ward and how they are treated/followed in this ward. Do they have

regular visits from clinical staff during their time in the virtual ward and how this type of care differs from what other jurisdictions may call hospital in the home.

Is the decision to discharge to a virtual ward influenced by the availability of other government or non government support services it easier to discharge to a Virtual ward if the patient has a partner or carer than those without. You mention that its easier to discharge to virtual care if the patient can self-care but is the availability of other carers such as a partner or other home services or patients residing in a nursing home able to provide care if a patient cannot self-care.

Thanks for this important point. We have added a new section explaining the Virtual Ward provision (page 5): *The Virtual Ward at Wrightington, Wigan, and Leigh Teaching Hospitals National Health Service Foundation Trust (WWL) initially originated as an oxygen service to facilitate the follow-up of COVID-19 patients. As the number of COVID-19 cases requiring follow-up decreased, the service underwent repurposing. The administration of virtual ward patient management utilised a software platform provided by Current Health, with monitoring equipment established by WWL medical electronics team to generate automated data through the Current Health web-based system. By the onset of 2022, the virtual ward had evolved into a stable service, providing a viable alternative to continuous hospital admissions.*

A dedicated team of core nurses maintained daily communication with patients through iPad/FaceTime. Patient data is systematically scrutinised through the analysis of automated observations within a cloud-based software framework. In instances where concerns arise about a patient's condition, virtual ward nurses coordinate a review with the responsible Consultant. In-person visits for tasks such as intravenous antibiotic administration, blood sample collection, or wound management were organised by the nurses. Continuous monitoring is conducted to track the patient's progress or detect signs of deterioration. In cases of deterioration, arrangements for readmission can be made, with transportation facilitated to the Emergency Department or Same Day Emergency Care through ambulance services if necessary.

The selection of patients for the virtual ward is a collaborative effort between medical and nursing teams on the ward. The virtual ward team conducts assessments to determine suitability, and upon confirmation, a technical setup visit is arranged at the patient's residence to verify self-sufficiency, assistance from a partner or family, or an existing care support system. Patients relocated to care homes are excluded from virtual ward consideration.

Discharge from the virtual ward is contingent upon assessments conducted by the virtual ward team and only when monitoring is deemed unnecessary for the patient. [Lines 101-122]

R1

Could you define or show show a reference to how the clinical frailty score is calculated is it based on ICD codes or a clinical assessment team that estimated the score

We have added this information in the Data section and provided relevant reference: *CFS is measured at triage by the care professional seeing the patient (it may also be reassessed at a later point in the admission by other healthcare professionals if further information becomes available). Attribution of one of the CFS score to a patient was carried with the support of the Clinical Frailty Scale App (Rockwood K, Theou O. Using the Clinical Frailty Scale in Allocating Scarce Health Care Resources. Can Geriatr J 2020;23(3):210-15. doi: 10.5770/cgj.23.463). [Lines 138-141]*

R1

How were comorbidities measured ie by scanning clinical notes or looking at secondary icd code?

Information added in the Data section: *ICD-10 based coding data was employed to assign comorbidities to each patient. Each patient's episodes of care were meticulously coded at the point of discharge, offering comprehensive insights into both the reason for admission and the documented co-morbidities. These co-morbidities were then cross-referenced with the records obtained at the time of admission to the virtual ward. In instances where disparities were identified between the recorded co-morbidities at admission and those during the episodes of care, the electronic notes of each patient were meticulously examined to establish the accurate set of co-morbidities. The electronic notes served as the reference in resolving any discrepancies, ensuring the utmost precision in our*

data interpretation. Due to the study design and rigorous data entry checks, there is no missing data.
[Lines 150-158]

R1

Would a calculated index such as Charlson or Elixhauser have been useful to describe comorbidities **This is an excellent point from the reviewer. The integration of the Charlson or Elixhauser co-morbidity index could have shed light on some of the variations in Virtual Ward (VW) admission, hospital re-admission, or patient survival. However, such inclusion might have obscured the specific effects of the co-morbidities under investigation, which constituted the primary focus of our study outcomes, as outlined in Table 1. To address this consideration, we have introduced the following paragraph in the discussions:** *The exclusion of co-morbidity indexes, such as the Charlson²⁵ or Elixhauser²⁶, in our study was driven by two considerations. Firstly, our research focused on the analysis of length of stay, survival, and readmission at the level of individual co-morbidities rather than a broader assessment of general co-morbidity status. Secondly, this approach allowed for a more detailed examination of specific health conditions and facilitated comparisons with findings from prior research that adopted a similar methodology. However, it is crucial to recognise that in any future research aimed at optimising the decision-making process regarding patients' admission to hospitals or virtual wards, an exploration of the discriminatory power of established co-morbidity indexes such as Charlson or Elixhauser would be highly beneficial.* [Lines 338-345]

Reviewer: 2

Dr. Chao-Jui Li, Chang Gung Memorial Hospital Kaohsiung Branch

R2

A case-control study is an observational research design used in epidemiology to investigate and compare individuals with a specific outcome or condition (cases) and individuals without that outcome or condition (controls). The primary aim is to identify factors or exposures that may be associated with the occurrence of the outcome.

In a case-control study, researchers select cases based on the presence of the outcome of interest and then choose controls without the outcome, often from the same population. The two groups are then compared with respect to various exposures or risk factors to assess their association with the outcome. Matching criteria, such as age, gender, and other relevant variables, may be employed to control for potential confounding variables and enhance the validity of the findings.

While this study performed substantial matching on various risk factors, the initial grouping was not based on a specific outcome but rather on differences in hospitalization patterns. I believe this does not align with the definition of a case-control study and is more akin to a retrospective cohort study

We understand the point of the reviewer, however other articles used the same definition (a non-clinical definition) of case-control studies, as for example:

- **Gaied, J., et al. (2022). "Virtual ward" community outreach support for COVID-19-positive hemodialysis patients may delay but not prevent subsequent admission to hospital: A single-center retrospective case-control pilot study.** *Hemodialysis International* 26(2): 278-280);
- **Svahn, Britt-Marie, Remberger, Mats, Heijbel, Mona, Martell, Eva, Wikström, Marie, Eriksson, Britta, Milovsavljevic, Ruza, Mattsson, Jonas, Ringdén, Olle. Case-Control Comparison of At-Home and Hospital Care for Allogeneic Hematopoietic Stem-Cell Transplantation: The Role of Oral Nutrition.** *Transplantation* 85(7):p 1000-1007, April 15, 2008. | DOI: 10.1097/TP.0b013e31816a3267

We acknowledge that the utilisation of non-clinical cases and controls (non-clinical due to attribution based on a non-clinical characteristic) is uncommon. In light of this, we have amended the study design to a 'retrospective longitudinal study' (see through the text).

R2

In addition, in the section on experimental design, there is insufficient description of the care differences experienced by patients admitted to the hospital and those admitted to the virtual ward. I believe a more detailed explanation should be provided.

As in response to R1 and R2, we have added a new section on the virtual ward provision:

(page 5): The Virtual Ward at Wrightington, Wigan, and Leigh Teaching Hospitals National Health Service Foundation Trust (WWL) initially originated as an oxygen service to facilitate the follow-up of COVID-19 patients. As the number of COVID-19 cases requiring follow-up decreased, the service underwent repurposing. The administration of virtual ward patient management utilised a software platform provided by Current Health, with monitoring equipment established by WWL medical electronics team to generate automated data through the Current Health web-based system. By the onset of 2022, the virtual ward had evolved into a stable service, providing a viable alternative to continuous hospital admissions.

A dedicated team of core nurses maintained daily communication with patients through iPad/FaceTime. Patient data is systematically scrutinised through the analysis of automated observations within a cloud-based software framework. In instances where concerns arise about a patient's condition, virtual ward nurses coordinate a review with the responsible Consultant. In-person visits for tasks such as intravenous antibiotic administration, blood sample collection, or wound management were organised by the nurses. Continuous monitoring is conducted to track the patient's progress or detect signs of deterioration. In cases of deterioration, arrangements for readmission can be made, with transportation facilitated to the Emergency Department or Same Day Emergency Care through ambulance services if necessary.

The selection of patients for the virtual ward is a collaborative effort between medical and nursing teams on the ward. The virtual ward team conducts assessments to determine suitability, and upon confirmation, a technical setup visit is arranged at the patient's residence to verify self-sufficiency, assistance from a partner or family, or an existing care support system. Patients relocated to care homes are excluded from virtual ward consideration.

Discharge from the virtual ward is contingent upon assessments conducted by the virtual ward team and only when monitoring is deemed unnecessary for the patient. [Lines 101-122]

VERSION 2 – REVIEW

REVIEWER	Hakendorf, P Flinders University
REVIEW RETURNED	07-Jan-2024
GENERAL COMMENTS	The authors have clarified points raised and made improvements to the original and I endorse this paper for publication